# Alleviating the Semantic Gap for Generalized fMRI-to-Image Reconstruction

**Tao Fang**[12], **Qian Zheng**[12]*, **and Gang Pan**[12]
[1]College of Computer Science and Technology, Zhejiang University, Hangzhou, China
[2]The State Key Lab of Brain-Machine Intelligence, Zhejiang University, Hangzhou, China
{duolafang, qianzheng, gpan}@zju.edu.cn

## Abstract

Although existing fMRI-to-image reconstruction methods could predict high-quality images, they do not explicitly consider the semantic gap between training and testing data, resulting in reconstruction with unstable and uncertain semantics. This paper addresses the problem of generalized fMRI-to-image reconstruction by explicitly alleviates the semantic gap. Specifically, we leverage the pre-trained CLIP model to map the training data to a compact feature representation, which essentially extends the sparse semantics of training data to dense ones, thus alleviating the semantic gap of the instances nearby known concepts (i.e., inside the training super-classes). Inspired by the robust low-level representation in fMRI data, which could help alleviate the semantic gap for instances that far from the known concepts (i.e., outside the training super-classes), we leverage structural information as a general cue to guide image reconstruction. Further, we quantify the semantic uncertainty based on probability density estimation and achieve **G**eneralized fMRI-to-image reconstruction by adaptively integrating **E**xpanded **S**emantics and **S**tructural information (**GESS**) within a diffusion process. Experimental results demonstrate that the proposed GESS model outperforms state-of-the-art methods, and we propose a generalized scenario split strategy to evaluate the advantage of GESS in closing the semantic gap. Our codes are available at https://github.com/duolala1/GESS.

## 1   Introduction

Functional magnetic resonance imaging (fMRI) is a powerful tool for studying the human brain and visual system, as it provides a non-invasive way to measure neural activity. Image reconstruction from fMRI data is important for studying visual representation in the cortex and for developing the vivid "reading the mind" brain-computer interface (BCI) technology [10, 19, 11].

High-quality fMRI-to-image reconstruction is a typical cross-modality problem [14, 28] and suffers from severe ill-posedness [2]. Existing state-of-the-art methods leverage the data-driven scheme to address such ill-posed problem by learning data prior from training data. However, training data are often collected as a limited number of instances [19], and real-world images are distributed in a wide, broad semantic space with a long-tail distribution[12]. This brings about the problem of the *semantic gap, the semantics of testing instances may be unknown in training stage*.

Addressing the semantic gap between the training data collected from the laboratory and the testing instances in the real world helps develop reconstructions for *generalized fMRI-to-image* scenarios and significantly promote the application of BCI. However, previous methods put too much attention on improving the image quality while less focus on the semantic accuracy of the reconstructed images, which brings two problems. **Unstable semantics**: The limited samples in each super-class fails to

---

*Corresponding author.

37th Conference on Neural Information Processing Systems (NeurIPS 2023).

form a compact feature space, which may result in incorrect decision boundaries and inability to estimate robust semantics even located within the known concepts (Fig.1c). This is what we called the inside-space gap (ISG). **Uncertain semantics**: The concepts covered by the training set is not enough, resulting in uncertainty in the prediction of test samples with unknown concepts (more like a zero-shot problem), which is called the outside-space gap (OSG). Traditional methods assume that the training set covers all the semantics in the test set (Fig.1b), and ignore the semantic gap caused by unknown samples in reality.

To this end, this paper addresses the generalized fMRI-to-image reconstruction problem by explicitly alleviating the semantic gap. To deal with the instances within known semantic space (ISG problem), we map the fMRI signals to a compact semantic space via a pre-trained Contrastive Language-Image Pre-Training (CLIP) model[23]. To deal with the instances within unknown semantic space (OSG problem), we propose to use the structure information as a transferable cue to guide the reconstruction, which is inspired by the robust and redundant low-level representation in visual cortex[19]. As it is difficult to find a hard boundary to define ISG and OSG cases for a given instance, we quantify its semantic confidence by probability density estimation on the training semantics and adopt the likelihood as the contribution indicator. Finally, we achieve **G**eneralized fMRI-to-image reconstruction by adaptively integrating **E**xpanded **S**emantics and **S**tructural information (**GESS**) in a diffusion process.

Our contributions in this paper could be summarized as:

- We explicitly address the generalized fMRI-to-image reconstruction problem and formulate its solution as alleviating the semantic gap within known and unknown semantic subspaces.

- We propose a CLIP based method to expand the fMRI features to a compact semantic space to alleviate the inside-space gap, and a structural information guided diffusion model to alleviate the outside-space gap.

- We construct a confidence indicator by quantifying the semantic similarity between a given instance and the training data, based on which we propose GESS to achieve generalized fMRI-to-image reconstruction by adaptively weighting the semantic and structural information.

- Our experimental results demonstrate that the proposed GESS model outperforms the classical and state-of-the-art methods. Additionally, we propose a dataset split method to construct a generalized fMRI-to-image scenario, which allows us to further evaluate the model's generalization ability.

## 2   Related works

### 2.1   Image reconstruction from fMRI

Decoding visual signals from the visual cortex has been investigated for a long time, ranging from early manual image feature extraction and linear mapping for prediction[20, 21] to later combining deep learning models (DNNs, GANs, DMs)[25, 15, 27, 10] and continuously improve the effect of generating images. Here we roughly divide the existing decoding methods into three categories:

(1) Reconstructing from the low-level features, such as [20] designs contrast patches as image stimuli and adopts linear mapping to predict from fMRI signals, [21] adopts Gabor features in reconstruction, and [2] directly establishes a bidirectional mapping between fMRI and images based on CycleGAN without explicitly modeling the semantics. Directly reconstructing the structural features is comparatively simple and efficient while suffers from the low quality results.

(2) Explicitly extracting high level conditions like categories for reconstructions. [6] refer to Masked Autoencoders (MAE)[16] to extract high-level features in a self-supervised manner, and [13] explicitly predicts category information by a classification task and uses it as conditions for a diffusion model. Such researches achieve vivid reconstructions due to the well-pretrained image generator, but have not considered the great semantic gap in reality. Obtaining incorrect semantics result in unrelated reconstructions that may look natural but meaningless.

(3) Hybrid methods that extract multiple-level features and jointly use them for reconstruction. [10] extracts high-level semantics and low-level shapes from different visual areas and adopt GAN for

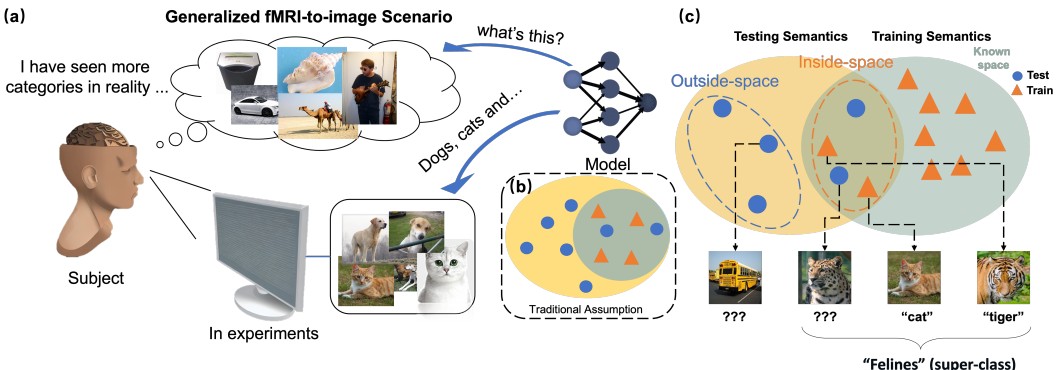

Figure 1: Illustration of the semantic gap in generalized fMRI-to-image reconstruction: (a) Humans have excellent generalization ability for long-tail distributed categories in the real world because they have seen a lot, but for models that have only seen limited fMRI data, it is difficult to generalize to more complex scenarios. (b) Traditional methods implicitly assume that the training set contains the semantics present in the test set. (c) We consider that in a generalized scenario, the training set forms a semantic space called the known space, and there is an intersection between the semantics in the testing set and the known space. In the inside-space scenario, although there may be unstable semantics, if mapped correctly, similar semantics (e.g., felines) can still be found in the neighborhood. However, in the outside-space scenario, the semantic divergence of the testing samples is too large, leading to uncertain or even irrelevant estimated semantics.

reconstruction. [27] also extracts high-level and low-level features as conditions for a diffusion model. [15] injects category information within a GAN reconstruction phase. The mentioned hybrid methods still do not achieve satisfactory results due to incorrect semantics caused by the semantic gap. We propose a hybrid method that not only explicitly extracts semantic and structural information, but also adaptively integrate the features with based on the semantic uncertainty, alleviating the semantic gap and achieving general and vivid reconstructions.

## 2.2 Large-scale pretrained models

With the development of large-scale models, [24, 8, 23] prove that models with a large enough number of parameters trained with large-scale datasets show more generality and practicality for many downstream tasks like zero-shot identification. **CLIP**[23, 30] is one pretrained model which predicts a wide range of visual concepts using natural language supervision. It proposes that learning directly from raw text about images provides a broader source of supervision, and scalably learn state-of-the-art image representations from scratch on a dataset of $400$ million. We obtain an image-text shared feature space through CLIP, where the point-wise distance indicates the potential semantic relationship, and implicitly provides an extending manifold for the limited training semantics and a potential interpolation ability in the prediction phase. **Diffusion models**[17] are one type of generative model that decompose the image formation process into a sequential denoising steps. The diffusion process destroies the image structure by gradually adding gaussian noise, and simulate the image generation as an inverse process using a UNet model[17]. Latent Diffusion Models (LDMs) [24] better handle high-resolution images with lower computational complexity in a latent diffusion process. We select LDM for image reconstruction due to the high-resolution and high-quality generation ability,

## 3 Methods

### 3.1 Problem Definition and Method Overview

Dividing the brain-image pair data as training $X^{\text{tr}}, Y^{\text{tr}}$ and testing set $X^{\text{te}}$ with the $Y^{\text{te}}$ are unknown, we aim to decode accurate and natural visual stimulus $\hat{Y}^{\text{te}}$ from the testing fMRI data $X^{\text{te}}$ and make $\hat{Y}^{\text{te}} \approx Y^{\text{te}}$. The visual features can be divided into two components: semantic information

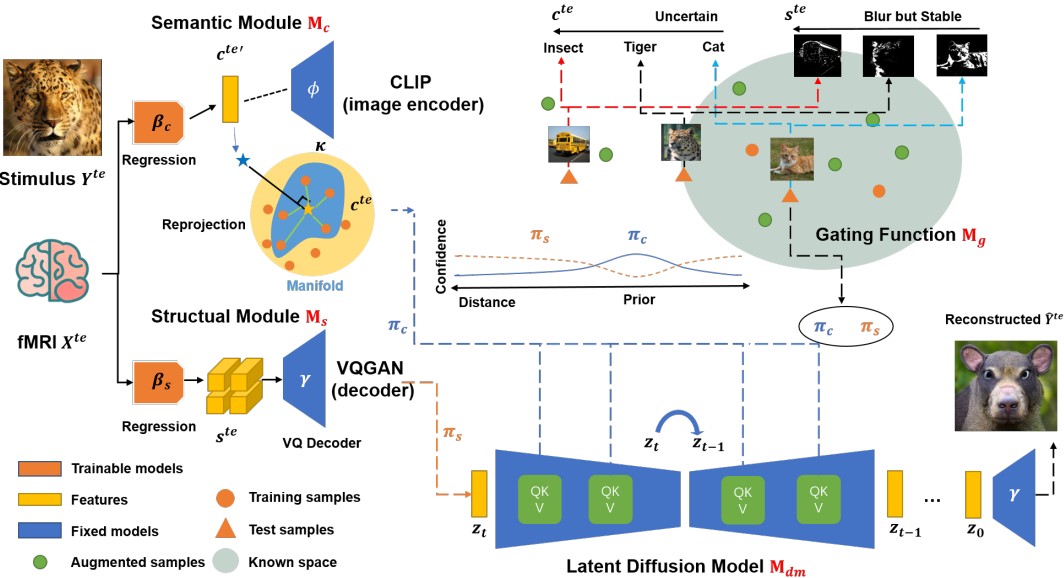

Figure 2: Framework of GESS. The semantic module $M_c$ maps fMRI data onto a semantic space using CLIP and then reprojects it onto the dense manifold to obtain faithful semantics $c^{te}$. Structural module $M_s$ extracts low-frequency features as structural information $s^{te}$. Gating function $M_g$ adaptively assigns $\pi_c$ for $c^{te}$ based on the semantic uncertainty, and assigns $\pi_s = 1 - \pi_c$ for $s^{te}$. LDM $M_{dm}$ reconstructs images conditioned on the two types of features weighted by $\pi_c, \pi_s$.

$c^{n \times d_c}$ and structural information $s^{n \times d_s}$ that shared between fMRI and images (where $n$ denotes the number of samples, $d_c, d_s$ denotes the feature dimension). Here $c$ is derived by a semantic module $M_c$ using CLIP (with parameter $\phi$), and $s$ is derived by a structural module $M_s$ using VQGAN (with parameter $\gamma$). To describe the generalized fMRI-to-Image scenario, letting $\mathcal{R}^d$ be the universal set of semantic space with $d$ dimension we get $c^{tr} \in \mathcal{R}^{tr}$ as the subspace of the training semantic features (that we call known space) and $c^{te} \in \mathcal{R}^{te}$ is the subspace of the testing semantic features and $\mathcal{R}^{tr} \subset \mathcal{R}^d, \mathcal{R}^{te} \subset \mathcal{R}^d$. As in Fig.1c, in the generalized fMRI-to-Image scenario, there is overlapping that $\mathcal{R}^{tr} \cap \mathcal{R}^{te} \neq \emptyset$ and $\mathcal{R}^{te} \not\subseteq \mathcal{R}^{tr}$ (different from traditional assumption that $\mathcal{R}^{te} \subseteq \mathcal{R}^{tr}$). To avoid confusion, from now on, when referring to space, it will be in the context of the training semantic space. Due to the different uncertainties of $c^{te} \in \mathcal{R}^{tr}$ and $c^{te} \notin \mathcal{R}^{tr}$ across samples caused by the generalized scenario, we introduce a gating function $\pi_c(c^{te})$ and $\pi_s(s^{te})$ in mixture of experts (MOE), to adaptively assign confidence for semantic and structural information. With the aforementioned components and their estimated weights, we obtain reconstructed images $Y^{te}$ from a conditioned latent diffusion model, $f_\theta(c^{te}, s^{te}, \pi_c, \pi_s)$.

In this paper we propose a two-stage decoding model: (1) To alleviate the semantic gap, we implement a semantic module ($M_c$ in Fig.2) that maps fMRI data onto a dense and continuous manifold constructed by CLIP and a structural module ($M_s$) to extract low-frequency features as a transferable clue. (2) To determine the ISG/OSG situation, we propose $M_g$ to estimate the inside-space likelihood to implicitly assess the situation and combine the two types of features using a weighted assignment approach in a diffusion process ($M_{dm}$).

## 3.2 Semantic gap alleviation

The visual concepts are long-tail distributed in the real world so that the semantics that collected in a limited BCI experiment could not cover enough real world concepts. To achieve reconstruction in the generalized fMRI-to-image scenario, we further divided the semantic gap into two cases, which have been addressed by modules $M_c, M_s$ respectively.

### 3.2.1 Semantic module $\mathrm{M_c}$: inside-space case

Even in the known space $\mathcal{R}^{\mathrm{tr}}$, there may not be enough samples to continuously represent the space due to the limited number of fMRI samples. Gaps in this subspace without instance filling may result in classification failure, even if a sample has similar concepts (super-classes) in the training set. This is referred to as the inside-space gap (ISG). As ISG is still a problem within the same domain, we propose that the fundamental reason lies in the discontinuous manifold directly constructed from the limited number of fMRI samples. As mentioned before, the CLIP feature space shows a locally continuous and semantically reasonable manifold (i.e., the cosine similarity reveals semantic relationships). By appropriately projecting the fMRI samples onto this manifold, it becomes possible to implicitly extend the discontinuous fMRI features into a manifold that covers semantics of both of the collected and unseen fMRI samples.

Concretely we use a linear model (ridge regression [18]) to predict $c$ from $x$ with large regularization $\lambda_c$ to limit the impact of noise on predicting semantics. We get $c^{\mathrm{tr}} = f_\phi(x^{\mathrm{tr}})$ and fit the coefficients $\beta_c$ of ridge regression with $\phi$ of CLIP fixed. Although linear models and regularization may reduce the diversity of semantics and make it difficult to capture non-linearity, the simple relation assumption shrinks the coefficient variance and make the model less prone to over-fitting to the noise.

However, the mean squared error (MSE) and ridge regression cannot guarantee that the estimated $c^{\mathrm{te}''}$ are mapped onto the low-dimensional manifold in the high-dimensional space. Therefore, we propose two methods to alleviate this problem.

**Momentum alignment**    We propose to align the statistics of the predictions and the image prior distributions to reduce semantic domain shift from a distributional perspective. To get the statistics of the manifold, we first sample $C^{\mathrm{tr}}$ from the manifold by feeding the training images into CLIP models. However, the scarcity of training samples makes it difficult to accurately describe the manifold. Here we use augmentation $C^{\mathrm{aug}} = f_\phi(X^{\mathrm{aug}})$ to help the manifold sampling (let $C^{\mathrm{all}} = C^{\mathrm{tr}} \cup C^{\mathrm{aug}}$) and to satisfy the local linearity to some extend. Previously we get $C^{\mathrm{te}''}$ by ridge regression $\beta_c$, and here we adjust its statistics by using a simple domain adaptation method named first-order momentum alignment like [13], which involves a whitening and an alignment process to get $C^{\mathrm{te}'}$ which has the same mean and variance as $C^{\mathrm{all}}$ (supplementary materials 6.2).

**Linear re-projection**    Here we directly projects $C^{\mathrm{te}'}$ onto the manifold. We assume that in a small local area of the manifold, there are enough feature points such that a manifold point can be approximated as a linearly weighted sum of its K-nearest neighbor feature points (Fig. 2). The projection principle is that we re-project the object point $c^{\mathrm{te}'}$ to $c^{\mathrm{te}}$ using its K-nearest points $c_k^{\mathrm{nb}} \subset \mathcal{K}$ on the manifold ($\mathcal{K} \subset C^{\mathrm{all}}$) by a linear combination model: $c^{\mathrm{te}} = \sum_{\mathcal{K}} w_k^{\mathrm{nb}} \cdot c_k^{\mathrm{nb}}$, where $w_k^{\mathrm{nb}}$ is the coefficients of each within-manifold points $c_k^{\mathrm{nb}}$. Here we solve a constrained least-squares problem:

$$\arg\min_{w_k^{\mathrm{nb}}} ||c^{\mathrm{te}} - w_k^{\mathrm{nb}} \cdot c_k^{\mathrm{nb}}||_2^2, \; s.t. \; w_k^{\mathrm{nb}} = 1, \tag{1}$$

to get the $c^{\mathrm{te}}$ that located in manifold for latter reconstruction (supplementary materials 6.1). This approach [5] essentially considers that the information that causes the components to deviate from the manifold is not important and could be removed.

### 3.2.2 Structural module $\mathrm{M_s}$: outside-space case

Although previous work has also considered extracting semantics [27], it has not addressed the extreme scenario where the image semantics are distributed outside the known space, which is hard to get trustworthy predictions. We call it outside-space gap (OSG). For a generalized scenario, we propose to use structural information $S^{\mathrm{te}}$ to compensate for the insufficient generalization caused by solely depending on semantics. That is because structural information is a category-independent attribute, and is robustly and redundantly represented in multiple cortex areas[19]. We use the low quality $S^{\mathrm{te}}$ to supplement the uncertain semantics in reconstructions.

To extract the low-frequency components as $S$, before fitting we conduct Gaussian filtering on visual stimuli $\mathrm{Gaussian}(Y^{\mathrm{tr}}, r)$ with a large kernel $r = 15$ to prevent the model from trying to extract the high-frequency components from fMRI. Besides this can be seen as to suppress the variance of the predictions and to some extent, suppress the noise. In this task a simple ridge regression could

already achieve acceptable $s^{\text{te}}$, which is fast and easy to implement. Concretely, we use a pretrained VQGAN to extract the ground truth representation $\hat{s}^{\text{tr}} = f_\gamma(y^{\text{tr}})$ of each image. Then, with a ridge regression $\beta_s$, we can establish the relationship between fMRI and the latent variables by minimizing $|f_{\beta_s}(X^{\text{tr}}) - \hat{S}^{\text{tr}} + \lambda\beta_s|$, where $\hat{S}^{\text{tr}}$ has been flattened here. Another approach is to use CycleGAN [2] that trained on $Y^{\text{tr}}$ to generate $S^{\text{te}}$, which shows higher quality. Since a large number of epochs are needed to fit the model, it is computationally more expensive, so that the above two methods represent a trade-off between quality and efficiency.

## 3.3 Adaptive Integration with LDM

### 3.3.1 Gating function $\text{M}_\text{g}$ for Integration

Although in the first part, we have explicitly divided the problem into the ISG and OSG sub-problems, and construct the faithful manifold combined with structural information to alleviate the semantic gap, but in reality it is difficult to determine the boundary between the ISG and OSG cases. As the boundary, probably determined by a threshold, would introduce an extra hyper-parameter which is inaccessible for the testset. Considering that different features have their own strengths and inspired by mixture of experts (MOE [3]), we regard $\text{M}_\text{c}(\text{x})$ and $\text{M}_\text{s}(\text{x})$ as two experts, and use a gating function $\pi_c(x), \pi_s(x)$ to indicate each expert's own regions within $X^{\text{te}}$. We assume that there exists an integrated intermediate information $h^{\text{te}}$ which could be marginalized as a weighted sum of:

$$p(h_i^{\text{te}}|x_i^{\text{te}}) = \pi_c(x_i^{\text{te}})p(c_i^{\text{te}}|x_i^{\text{te}}) + \pi_s(x^{\text{te}})p(s_i^{\text{te}}|x_i^{\text{te}}), s.t. \pi_s + \pi_c = 1, \tag{2}$$

where $i = \{1, ..., n\}$, $\pi_s$ and $\pi_c$ are gated functions of the semantic and structural modules which works as adaptive weights for each component, and $p_c(c_i^{\text{te}}|x_i^{\text{te}}), p_s(s_i^{\text{te}}|x_i^{\text{te}})$ describes the intermediate features derived from the two modules respectively.

Here we propose to estimate the gating function in the form of likelihood of inside known space as $\pi_c(x_i^{\text{te}}) = p(c_i^{\text{te}}|C^{\text{tr}})$ and compute $\pi_s(x_i^{\text{te}}) = 1 - \pi_c(x_i^{\text{te}})$. This could be interpreted as samples that are far away from the manifold that the training semantics covered are more uncertain. Concretely, we propose a non-parametric density estimation method based on kernel density estimation (KDE[3] with bandwith=1.5, and then normalized to $[0, 1]$), which uses training samples to estimate the underlying probability distribution to approximately estimate $p(c_i^{\text{te}})$ on $C^{\text{tr}}$ (supplementary materials 6.3).

### 3.3.2 Diffusion model $\text{M}_{\text{dm}}$ for generation

Considering the powerful generality of LDM, we propose to implement MOE within the diffusion process. But directly taking the conditions in the form of cross attention inputs is not appropriate as it is difficult to realize feature weighting and a structural guidance containing spatial information. For MOE model, we need a common space to integrate the different modalities of $c^{\text{te}}$ and $s^{\text{te}}$. We first consider weighting in the gradient space and introduce a gradient guided method (GG for short), and inspired by [8] we get:

$$p_{\theta,\phi}(z_{i,t}|z_{i,t+1}, h_i) = \text{A}\, p_\theta(z_{i,t}|z_{i,t+1})\, p_\eta(h_i|z_{i,t}), \tag{3}$$

where A is the normalization term, $\theta, \eta$ denote the parameters of the UNet and the guidance module (like a classifier), $z_{i,t}$ means the latent features of sample $i$ at time step $t$ in the LDM. For the sake of readability, we omit the '$i$' and 'te' tags in the following. As mentioned before we have estimated the confidence $\pi_c$ and $\pi_s$ from $x_i$ and get $p_\eta(h|z_t) = \pi_c p_c(c|z_t) + \pi_s p_s(s|z_t)$ with the confidence fixed during the diffusion process. By Jensen's inequality we obtain the lower bound of log likelihood:

$$\log(p_\theta(z_t|z_{t+1})p_\phi(h|z_t)) = \log(p_\theta(z_t|z_{t+1})) + \log \sum_m \pi_m p_m(m|z_t) \tag{4}$$

$$\geq \log(p_\theta(z_t|z_{t+1})) + \sum_m \pi_m \log p_m(h_m|z_t), m \in \{c, s\}, \tag{5}$$

maximizing which across the diffusion process is equivalent to maximizing the log likelihood, in this way we realize the instance-wise adaptive weighting of different image features in the diffusion process. Further with Taylor expansion as in [8], we get an approximation of the likelihood:

$$\log p_m(h_m|z_t) \approx (z_t - \mu_t)g_m + \text{B}, m \in \{c, s\}, \tag{6}$$

wherein the gradient information $g_i$ comes from a loss term in the guidance module $\eta$ that measure accuracy of a certain decoded feature, and B is a constant term. In this case the $g_{i=c}$ comes from guidance of CLIP features $C^{\mathrm{te}}$ and $g_{i=s}$ comes from $S_t$ with a $\mathcal{L}_{\mathrm{mse}}$ guidance. Known that the latents $z$ means a perceptual compression[24, 9] of raw image, it still retains certain image properties that performing image manipulations such as kernel smoothing in the $z$-space makes sense. Here we apply low-pass filter $\mathrm{F}(z)$ to extract structural information from $z_t$ and get $\mathcal{L}_{\mathrm{mse}} = \| \mathrm{F}(z_0') - \mathrm{F}(s) \|_2^2$, where $z_0'$ are estimated from $z_t$[26]. One advantage of gradient guidance lies in its flexibility, like combination of image variance loss and other optimizations to improve the quality of the generated images.

However, GG method requires computing the gradients at each timestep, which greatly increases computational complexity. Inspired by ILVR [7], which can directly replace different frequency components in the image-space, we propose an efficient component substitution method (CS for short). Firstly we propose the component segmentation method:

$$z_t = \sum_m \mathrm{F}_m(z_t), m \in \{c, s\}, \tag{7}$$

where $\mathrm{F}_m$ means the filtering functions for different levels of information (e.g., high or low frequency information). In this work we have two-level features: semantics $\mathrm{F}_c$ and structure $\mathrm{F}_s$. $\mathrm{F}_s(z_t)$ can be obtained through low-pass filtering function and get $\mathrm{F}_c(z_t) = z_t - \mathrm{F}_s(z_t)$. We compute each frequency component of $z_t$ as a weighted combination of various experts: $\mathrm{F}_m(z_t) = \pi_s \mathrm{F}_m(z_{t,s}) + \pi_c \mathrm{F}_m(z_{t,c})$, where $z_{t,c}, z_{t,s}$ means the latent features derived from the $c, s$ conditions respectively. However, for the structural guidance $\mathrm{F}_c(z_{t,s})$ does not contain accurate semantics that the $\pi_c \mathrm{F}_c(z_{t,s})$ term does not make sense. Thus we only focus on the low-frequency component here. Referring to [7] we get $z_{t-1}$ from $z_t$:

$$z_{t-1,c} = \sim p_\theta(z_{t-1}|z_t, c), \tag{8}$$
$$z_{t-1} = z_{t-1,c} - \pi_s \mathrm{F}_s(z_{t-1,c}) + \pi_s \mathrm{F}_s(z_{t-1,s}), \tag{9}$$

where we compute $z_{t-1,s}$ by gradually adding noise in $s$ that has been estimated by $\mathrm{M}_s$. Since the generation process does not require gradient estimation, it is faster and requires less computational resources compared to the previous method. However, the generated images sometimes become blurry (possibly due to the discontinuous low-level substitution that may disrupt the image structure). Therefore, the two methods are a trade-off between efficiency and image quality.

## 3.4 Training details

In our model only the ridge regression modules of $\mathrm{M}_s, \mathrm{M}_c$ need to be trained, while the other modules are pretrained on large-scale data with fixed parameters. The overall process of our method is described in Algorithm 1.

**Semantic information**. To extract semantic features, we use the pretrained VIT-L/14 clip model on 1200 training images and about 4000 augmentation images. We use ridge regression trained with $\lambda_c = 1000$. We slightly smooth the images with a Gaussian kernel ($r = 5$) before extracting semantic features.

**Structural information**. To extract structural features, images are pre-smoothed with a Gaussian kernel ($r = 15$). In experiments we use CycleGAN with the same setting as [2]. We do not use augmentation as [2] because high-quality details are not necessary here.

**Generating**. We adopt the pretrained LDM in [4](knn2img model). During integration, both of the two methods mentioned in section 3.3 are implemented and compared in section 4.4, and we finally select CS method considering its efficiency and acceptable generation. A temporally scaling strategy is used in diffusion process to reduce the influence of $z_{t,s}$ and results in clearer generated images. We use DDIM [26] acceleration during sampling with 50 time steps.

**Algorithm 1** The pseudo code of GESS.

---

**Input**: Paired fMRI $X$ and Image $Y$ Dataset: $\mathcal{D}^{\text{tr}} = \{(x_i^{\text{tr}}, y_i^{\text{tr}})\}_{i=1}^n$ and $\mathcal{D}^{\text{te}} = \{(x_i^{\text{te}})\}_{i=1}^n$.

**Output**: Reconstructed stimulus images $\hat{y}^{\text{te}}$ from fMRI $x^{\text{te}}$.

**Training**:

Initialization: Constructing CLIP [23], VQGAN [9] and latent diffusion model [24] by pretrained parameters $\phi$, $\gamma$ and $\theta$.

Training parameter $\beta_c$ of semantic module $M_c$:

1. Extracting semantic features $\hat{c}_i^{\text{tr}}$ from $y_i^{\text{tr}}$ by CLIP: $\hat{c}_i^{\text{tr}} = f_\phi(y_i^{\text{tr}})$.

2. Training ridge regression parameters $\beta_c$ by $\hat{c}_i^{\text{tr}}$ and $x_i^{\text{tr}}$: $\beta_c = (\hat{C}^T \hat{C} + \lambda I)^{-1} \hat{C}^T X^{\text{tr}}$, where $\hat{C} = [\hat{c}_1^{\text{tr}}, \hat{c}_2^{\text{tr}}, ..., \hat{c}_N^{\text{tr}}]$ and $X^{\text{tr}} = [x_1^{\text{tr}}, x_2^{\text{tr}}, ..., x_N^{\text{tr}}]$.

Training parameter $\beta_s$ of structural model $M_s$:

1. Extracting structural features $\hat{s}_i^{\text{tr}}$ from $x_i^{\text{tr}}$ by VQGAN: $\hat{s}_i^{\text{tr}} = f_\gamma(y_i^{\text{tr}})$.

2. Flatting $\hat{s}_i^{\text{tr}}$, and training ridge regression parameters $\beta_s$ by $\hat{s}_i^{\text{tr}}$ and $x_i^{\text{tr}}$: $\beta_s = (\hat{S}^T \hat{S} + \lambda I)^{-1} \hat{S}^T X^{\text{tr}}$, where $\hat{S} = [\hat{s}_1^{\text{tr}}, \hat{s}_2^{\text{tr}}, ..., \hat{s}_N^{\text{tr}}]$.

**Inference**:

1. **Semantic module** Predicting semantics from fMRI: $c_i^{\text{te}''} = \beta_c x_i^{\text{te}}$. Using momentum alignment to get $c^{\text{te}'}$ (section 3.2.1). Using linear re-projection to get $c_i^{\text{te}}$ (section 3.2.1).

2. Predicting structure from fMRI: $s_i^{\text{te}} = \beta_s x_i^{\text{te}}$.

3. **MOE** Estimating weighting parameters $\pi_c$ by KDE and $\pi_s = 1 - \pi_c$ (section 3.3.1).

4. Latent diffusion process by CS strategy (section 3.3.2):

   (a) Initializing $z_0$ by Gaussian.

   (b) For $t = 1, 2, ..., T$:
   
       i. Conditioned by cross attention: $z_{t-1,c} \sim p_\theta(z_{t-1} | z_t, c^{\text{te}})$
   
       ii. Conditioned by CS strategy: $z_{t-1} = z_{t-1,c} - \pi_s F_s(z_{t-1,c}) + \pi_s F_s(z_{t-1,s})$

   (c) $y_i^{\text{te}} = f\gamma(z_T)$.

---

# 4 Results

## 4.1 Dataset

We evaluated the performance of our model using two datasets: the General Object Decoding (GOD) dataset [19] and the Natural Scenes Dataset (NSD)[1]. The GOD dataset contains 1200 images from 150 categories for training and 50 images from 50 categories for testing. NSD uses images from the COCO dataset and roughly 10,000 fMRI-image pairs for one subject. During a visual experiment, researchers recorded subjects' brain activities by fMRI scanners while they are watching images on a screen. In GOD, there was no overlapping between the training and testing classes which matches the generalized scenario (supplementary materials 7.1).

However, the random split method used in [27] for NSD causes too much overlap between the training and testing semantics (Fig.4a left), so that a simple baseline model (Fig.4a left) by nearest searching can achieve comparable performance. To address this issue, we proposed a generalized split strategy for NSD in Section 4.4. We showed that while the simple baseline model perform worse under this strategy, our model is still able to achieve stable reconstruction in this situation.

## 4.2 Evaluation of image similarity

Comparing and estimating image similarity is a challenging task in computer vision as several factors like object dislocation, different viewpoints greatly increases pixel-wise distance like L2. To measure the similarity between the reconstructed and ground truth visual stimulus, we propose a combined comparison criteria that builds upon previous works with slight modifications. Specifically, we use: (1) a 2-way comparison task as in [19] with three different metrics, including RMSE, Spectral Angle Mapper(SAM[29], a physically-based metric), and a perceptual similarity metric based on a

pretrained CLIP (VIT-L/14), to make an overall assessment of reconstruction accuracy, and (2) visual comparison of the reconstructed images from different methods. Besides to remove the randomness caused by pairwise comparison, we repeat and averaged 10 trials for each sample.

We compare our method with several existing methods on GOD: GAN-based methods include [2], which use CycleGAN to directly decode images from fMRI data with augmentation and [22] use ICGAN as backbones. Diffusion-based methods include [27] use MAE to extract features unsupervisedly as conditions and inputs into a latent diffusion model, and [13] predict image labels from fMRI and use a diffusion model to reconstruct the image conditioned on the category information. Most previous methods approach LDM as a purely generative model without exercising tight control over the diffusion process.

## 4.3 Performance comparison

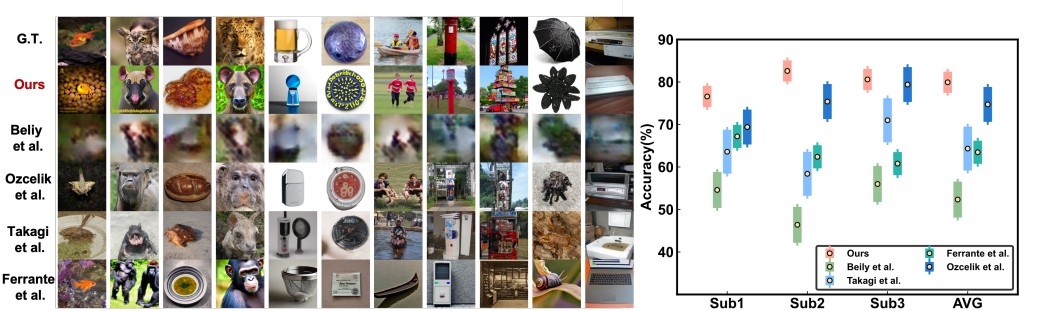

Figure 3: Performance comparison. Left: visual comparison of reconstructed images from different methods (more examples could be found in supplementary materials). Right: quantitative comparison of different methods by perceptual similarity.

We performed a quantitative and visual comparison between our model and several candidate models using the GOD dataset on three subjects ($1 \sim 3$). We used $1200$ and $50$ samples as the training and testing samples, respectively. Each sample has been averaged between $5$ and $24$ samples to improve signal-to-noise ratio. We also randomly sampled around $4000$ images from ImageNet for augmentation. The final results are shown visually and quantitatively in Fig.3. We outperform the recent published methods [22, 27] by a large margin (3.8%, 13.8% by perceptual similarity, 7.2%, 5.3% by SAM, 1.1%, 16.3% by RMSE). As can be seen in Fig.3 left we generate high-quality and high-resolution ($768 \times 768$) images that are more realistic in terms of their content than others. By explicitly taking the generalized scenario into account, our method outperforms the other methods visually and quantitatively.

## 4.4 Ablation study

Table 1: Abalation study on GOD dataset (Perceptual similarity).

| Subjects | $c^{\text{te}}$+LDM | $c^{\text{te}}$+$s^{\text{te}}$+CS | $c^{\text{te}}$+$s^{\text{te}}$+MOE+CS | $c^{\text{te}}$+$s^{\text{te}}$+MOE+GG |
|---|---|---|---|---|
| Sub1 | 74.2% | 69.2% | **78.0%** | 76.2% |
| Sub2 | 77.6% | 75.2% | **84.8%** | 76.6% |
| Sub3 | 79.6% | 77.2% | 80.4% | **82.6%** |
| AVG | 77.1% | 73.9% | **81.1%** | 78.4% |

To evaluate the usefulness of the components in our approach, we conducted an ablation study on the GOD dataset. We compared the performance of different combinations of our components, mainly to evaluate the influence of semantic and structural information on reconstruction, and the importance of MOE. As shown in Table 1, equal-weighted feature integration without considering the semantic uncertainty may even lead to negative optimization. When the structural information is weighted as in columns 4 and 5, it can improve reconstruction accuracy by 4% and 1.3% on average compared to using only semantics, respectively. It can also be observed that the CS method achieves higher accuracy than the GG method, even though the latter produces more natural-looking images. In the

supplementary materials we provide detailed evaluations of both the individual effectiveness of each component and the associated computational costs (in Table 4 and Table 5).

## 4.5 A generalized split strategy

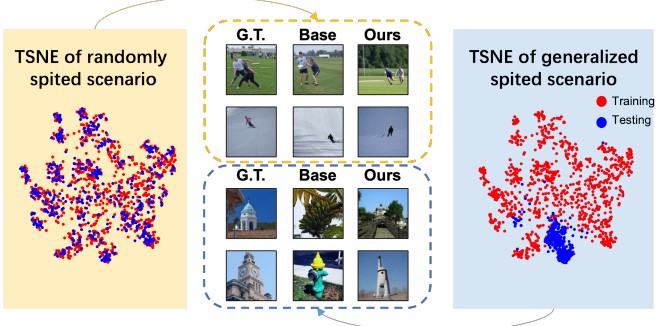

| Method | SAM | RMSE | Perceptual |
|--------|------|------|-----------|
| base+RS | 60.8% | 62.3% | 81.0% |
| base+GS | 50.8% | 48.5% | 52.0% |
| ours+RS | 66.8% | 63.5% | 88.8% |
| ours+GS | 60.5% | 64.5% | 63.3% |

Figure 4: Left: visualize the difference of scenario in semantic space. Our model achieves stable reconstructions across scenarios visually. Right: quantitative results of baseline model and our model on different scenarios. RS:random scenario;GS:generalized scenario.

We emphasize that the generalized fMRI-to-Image setting is crucial in forcing the model to learn the visual mechanism of the brain, instead of relying on a pattern matching strategy that simply retrieves from the database. Here we propose a simple generalized scenario split strategy on NSD. We first encode the images into a semantic space and then cluster the images into several clusters (in our case 20 clusters) by K-means to obtain pseudo labels. Then we construct a graph with the cluster centers as nodes and their cosine similarity as edges, and apply a minimum cut algorithm to split the graph into two parts with the minimum number of disconnected edges. As visualized in TSNE in Fig.4 middle, a comparatively larger concept gap exists between the training and testing sets in the semantic space.

As in Fig.4, we propose a simple retrieval model based on k-nearest-neighbor (KNN) on NSD as a baseline model (supplementary materials 7.2), which can generate relatively accurate reconstructions based on $1_{st}$ neighbor on the dataset that has been randomly split as [27] (Fig.4 left). Its performance deteriorates in a generalized experimental setting (Fig.4 right). Here we demonstrate that in such generalized scenario, our method still perform more stable reconstructions, even though the reconstruction accuracy has decreased by 28.4% compared to the baseline model (37.0%).

## 4.6 Disscussion

In this paper, we explicitly define the generalized fMRI-to-image scenarios and the great semantic gaps that they bring. To explicitly address the semantic gap, we decompose it into inside-space and outside-space cases and propose a model called GESS (**G**eneralized fMRI-to-image reconstruction by adaptively integrating **E**xpanded **S**emantics and **S**tructural information) that adaptively integrates semantic and structural information within a diffusion process. Our experimental results demonstrate that GESS achieves state-of-the-art performance.

**Limitations.** Our model still suffers from several limitations including: (1) The significant noise exists in fMRI signals, which may cause variability in the quality of generated images. This is a common challenge in fMRI-based visual reconstruction and requires further research to improve robustness. (2) Sampling one image over multiple time steps in LDM still has greater computational cost compared to GANs. (3) The current model needs retraining for each specific subject and cannot handle cross-subject data. As collecting experimental recordings is expensive, designing a model capable of cross-subject prediction could reduce the difficulty of applying it on new subjects.

## 5 Acknowledgments

This work was supported by STI 2030 Major Projects (2021ZD0200400), Natural Science Foundation of China (No. 61925603, U1909202), the Key Research and Development Program of Zhejiang Province in China (2020C03004).

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
