# Supplementary Materials for "Alleviating the Semantic Gap for Generalized fMRI-to-Image Reconstruction"

**Tao Fang**[12], **Qian Zheng**[12],[*] **Gang Pan**[12]
[1]College of Computer Science and Technology, Zhejiang University
[2]The State Key Lab of Brain-Machine Intelligence, Zhejiang University
{duolafang, qianzheng, gpan}@zju.edu.cn

## 5 Broader impacts

In this paper, we propose a novel approach, Generalized fMRI-to-image reconstruction by adaptively integrating Expanded Semantic manifold and Structural information (GESS), to address the semantic gap in the fMRI-to-image reconstruction problem. The improvement in fMRI-to-image reconstruction can lead to a better understanding of the human visual system and the neural representations of visual stimuli, thus significantly enhancing the potential applications of brain-computer interface (BCI) technologies. However, it is essential to consider the ethical implications of such advancements. As BCI technologies move closer to being able to "read the mind", privacy and consent concerns may arise. It will be crucial to develop policies and guidelines for the responsible use of these technologies and ensure that they are employed in a manner that respects individuals' rights and autonomy.

## 6 Details of methods

### 6.1 Reprojection details

The proposed method [4] aims to refine the extracted semantic components by finding their closest plausible counterparts in a given manifold. The method involves first finding the K nearest samples ($c_k^{nb} \in \mathcal{K}$) in the dataset under the cosine distance and then seeking a linear combination of these neighbors to reconstruct the semantic component $c^{te'}$ by minimizing the reconstruction error. The interpolation weights can be found by solving a constrained least-squares problem. Once the weights are solved, the reprojected point $c^{te'}$ on the underlying manifold can be computed.

To project the feature vector of $c^{te'}$ component to its manifold, the method assumes that the underlying manifolds are locally linear. The main idea of the classic locally linear embedding (LLE) algorithm is followed to achieve this. As mentioned in the main text, we use equation (1) to solve an optimization problem for finding the weights, which can then be used to calculate the projected point $c^{te'}$:

$$c^{te'} = \sum_{k \in \mathcal{K}} w_k^{nb} c_k^{nb}, \tag{10}$$

In practice, we concatenate the semantic vectors of $c^{te}$ and $c^{te'}$ together as the conditional input to LDM.

---

[*]Corresponding author.

37th Conference on Neural Information Processing Systems (NeurIPS 2023).

## 6.2 Domain adaptation details

A first-order domain alignment method aims to align the mean and variance of the feature distributions between the source and target domains. This can be achieved by transforming the features in such a way that the statistical properties of the source and target domains become similar.

First, we need to whiten the testing set. Whitening is a preprocessing technique used to remove correlations in the data and make it have a unit variance. Given a testing set $C^{te}$ we can whiten it using the following equation:

$$C_{whitened} = \frac{C^{te} - \mu^{te}}{\sigma^{te}} \tag{11}$$

where $\mu^{te}$ and $\sigma^{te}$ are the mean and standard deviation of the testing set respectively.Next, we need to align the mean and variance of the testing set to the training set. This can be achieved using the following equation:

$$C_{aligned} = \sigma^{tr} \cdot C_{whitened} + \mu^{tr} \tag{12}$$

where $\mu^{tr}$ and $\sigma^{tr}$ are the mean and standard deviation of the training set respectively.In this process, we first whiten the testing set, removing correlations in the data and making it have unit variance. Then, we align the mean and variance of the testing set to the training set, making the training set and testing set have similar statistical properties.

## 6.3 Non-parametric kernel density estimation in gating function

To estimate the likelihood of $c^{te}$ being sampled from $c^{tr}$, we use a non-parametric kernel density estimation (KDE) method for $p(c^{te}|C^{tr})$. KDE[3] is a non-parametric method for estimating the probability density function of a random variable, based on a set of observations. It is a widely used technique in statistical analysis and machine learning, as it does not require any assumptions about the underlying distribution of the data.

KDE works by placing a kernel function at each observation point and summing up the contributions from all kernels to estimate the density at any given point. The kernel function is typically a probability density function itself, centered at the observation point and with a bandwidth parameter that determines the width of the kernel.In this case we select Gaussian kernel, which has the form:

$$K(d_c) = \frac{1}{\sqrt{2\pi} \, b_w} \; e^{-\frac{1}{2}\left(\frac{d_c}{b_w}\right)^2}, \tag{13}$$

where $d_c$ denotes the feature distance to be measured and $b_w$ is the bandwidth parameter, controlling the width of the kernel (1.5 in our case). The estimated density function is then given by:

$$\hat{f}(c^{te}) = \frac{1}{n \cdot b_w} \sum_{i=1}^{n} K\left(\frac{c^{te} - c^{tr}(i)}{b_w}\right) \tag{14}$$

where $n$ is the number of $C^{tr}$ and $x_i$ is the $i$-th observation. Note that choosing the appropriate bandwidth parameter can be challenging, as a too small value can result in over-fitting and a too large value can result in under-fitting. In our method cross-validation has been used to select an optimal value for $b_w$.

# 7 Details of experiments

## 7.1 More details of the datasets

**GOD dataset.** The God dataset[8] contains fMRI data collected from five healthy subjects during visual image presentation and imagery experiments. The subjects are one female and four males, aged 23 to 38 years with normal or corrected-to-normal vision. A sample size of five subjects was chosen to match previous fMRI studies with similar behavioral protocols. All subjects are highly experienced with participating in fMRI experiments and provided written informed consent. The study was approved by the Ethics Committee of ATR. For the visual image presentation experiment, images were selected from 200 object categories in the ImageNet database. After excluding low-quality images, the remaining images were cropped to the center. Subjects viewed the images in 24 training

runs and 35 test runs while maintaining central fixation. In each 9-second stimulus block, 12 images of the same category were flashed at 2 Hz. Subjects performed a one-back repetition detection task to maintain attention. The training runs contained 1200 images from 150 categories, with 8 images per category. The test runs contained 50 images from 50 new categories, each presented 35 times. In this paper we used data of three subjects and pre-process the data in the same way as [2].

**NSD dataset** As in [10], we utilized the Natural Scenes Dataset which provides fMRI data from subjects viewing repeated presentations of natural images. In [10] they analyzed data from 4 out of 8 subjects who completed all scans. For the random split scenario in section 4.5, we used 982 images as testing set which were viewed by all subjects. The remaining 24980 trials were used for training following [10], and more descriptions could be found in [1].

**Pairwise comparison** In accordance with [5], we assessed the semantic accuracy of our results using the n-way classification task. This involved performing multiple trials (10 trials in our tasks) and calculating the top one classification accuracy for $n-1$ randomly selected classes plus the correct one. It is worth noting that we take into account both of the pixel-level metrics (RMSE and SAM) and the semantic-level metrics (perceptual similarity), as objective in this work is to recover images with both structurally and semantically correct features.

## 7.2 The implementation details of the baseline method

In section 4.5 we propose that in the random split scenario [10], the model only has to learn a retrieval based model (i.g., a K-nearest neighbor model) to achieve a good performance without learning visual mechanism. Here we propose a simple KNN model as a baseline model to achieve comparatively good performance based on the random split.

Here we implement a ridge regression method (parameter $\beta_n$) to predict the semantic vector of the test image $c^{te}$ from $x^{te}$. The $\beta_n$ is fitted to learn the mapping from $X^{tr}$ to $C^{tr}$ that has been extracted by a pretrained and fixed CLIP model. Then the predicted $c^{te}$ is used to retrieve the most similar image in $X^{tr}$ as reconstructions by calculating the cosine distance between $c^{te}$ and $C^{tr}$.

Pairwise comparison experiments show that while simple, this model achieves comparable performance on the random split method (81.0% accuracy by perceptual similarity). More 1st nearest neighbor images are shown in Figure 5.

Table 2: Pairwise comparison results of SAM similarity on GOD.

| Subject | Sub1 | Sub2 | Sub3 | AVG |
|---|---|---|---|---|
| Ours | **56.8%** | **58.6%** | **66.6%** | **60.7%** |
| Beliy et al.[2] | 49.2% | 51.2% | 55.4% | 51.93% |
| Ozcelik et al.[9] | 53.2% | 52.4% | 54.6% | 53.4% |
| Takagi et al.[10] | 53.8% | 54.8% | 47.8% | 52.1% |
| Ferrante et al.[7] | 53.4% | 55.0% | 57.6% | 55.3% |

Table 3: Pairwise comparison results of RMSE similarity on GOD.

| Subject | Sub1 | Sub2 | Sub3 | AVG |
|---|---|---|---|---|
| Ours | **67.6%** | 64.8% | 66.6% | 68.1% |
| Beliy et al.[2] | 62.2% | **66.8%** | **76.4%** | **68.5%** |
| Ozcelik et al.[9] | 55.4% | 54.6% | 54.6% | 55.0% |
| Takagi et al.[10] | 50.8% | 52.6% | 47.8% | 51.7% |
| Ferrante et al.[6] | 65.6% | 65.2% | 57.6% | 67.1% |

## 7.3 Additional results of performance comparison

In addition to the experimental results presented in the main text, Tables 2 and 3 list the accuracy of various methods in pairwise comparison experiments based on RMSE and SAM similarity metrics. By RMSE similarity, [2] performs better than other diffusion-based methods, which is probably because the metric prefers smooth and blurry images.We also present the component substitution-based reconstruction results of our method for all 50 test images across three subjects in Figures

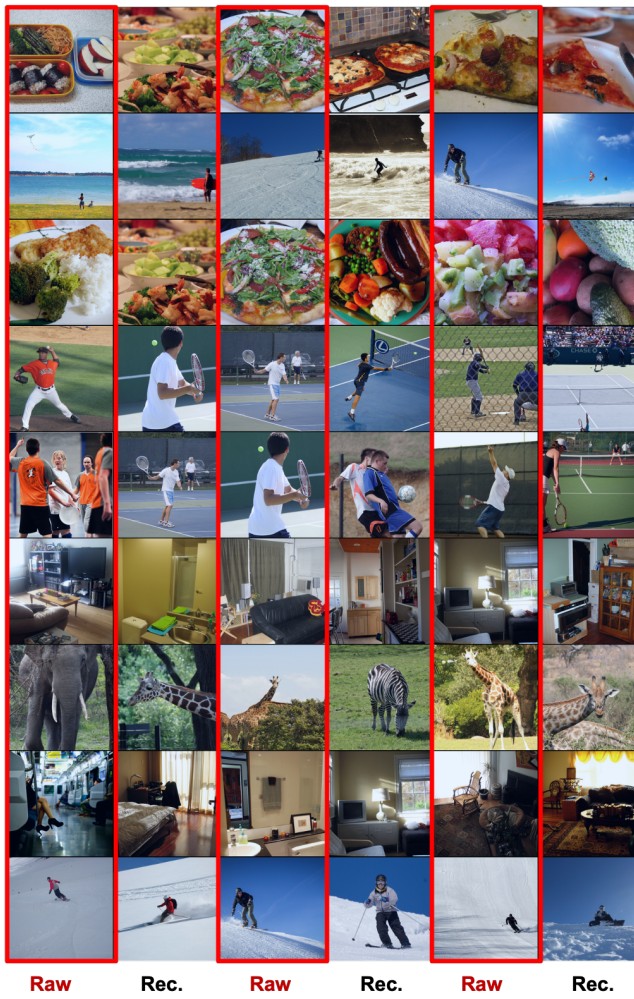

**Raw**    Rec.    **Raw**    Rec.    **Raw**    Rec.

Figure 5: More baseline method's reconstructions in the random split scenario.

Table 4: Time consuming of each module.

| Method | Time Consuming |
|---|---|
| Gradient Guidance | 679.87 s |
| Component Substitute | 215.45 s |
| Fitting Semantic Module $M_c$ | 0.158 s |
| Fitting Structual Module $M_s$ | 2.459 s |
| CLIP embedding (pre-processing) | 47.553 s |
| VQVAE embedding (pre-processing) | 72.366 s |

6-8. The results indicate that the reconstruction quality has some slight differences across different subjects, but our method can achieve good reconstructions for all. The results shown in the main text are from subject 3.

# References

[1] Emily J Allen, Ghislain St-Yves, Yihan Wu, Jesse L Breedlove, Jacob S Prince, Logan T Dowdle, Matthias Nau, Brad Caron, Franco Pestilli, Ian Charest, et al. A massive 7t fmri dataset to bridge cognitive neuroscience and artificial intelligence. *Nature neuroscience*, 25(1):116–126, 2022.

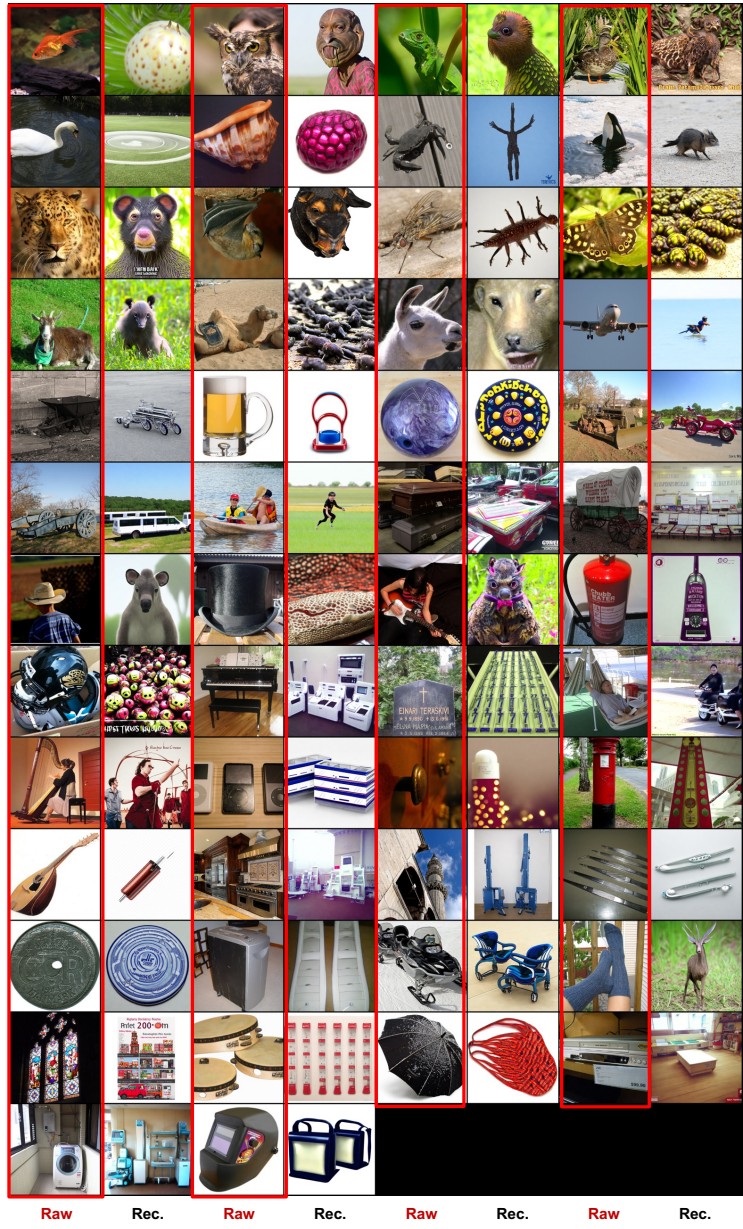

Figure 6: Fully sampled and reconstructed images of subject 1.

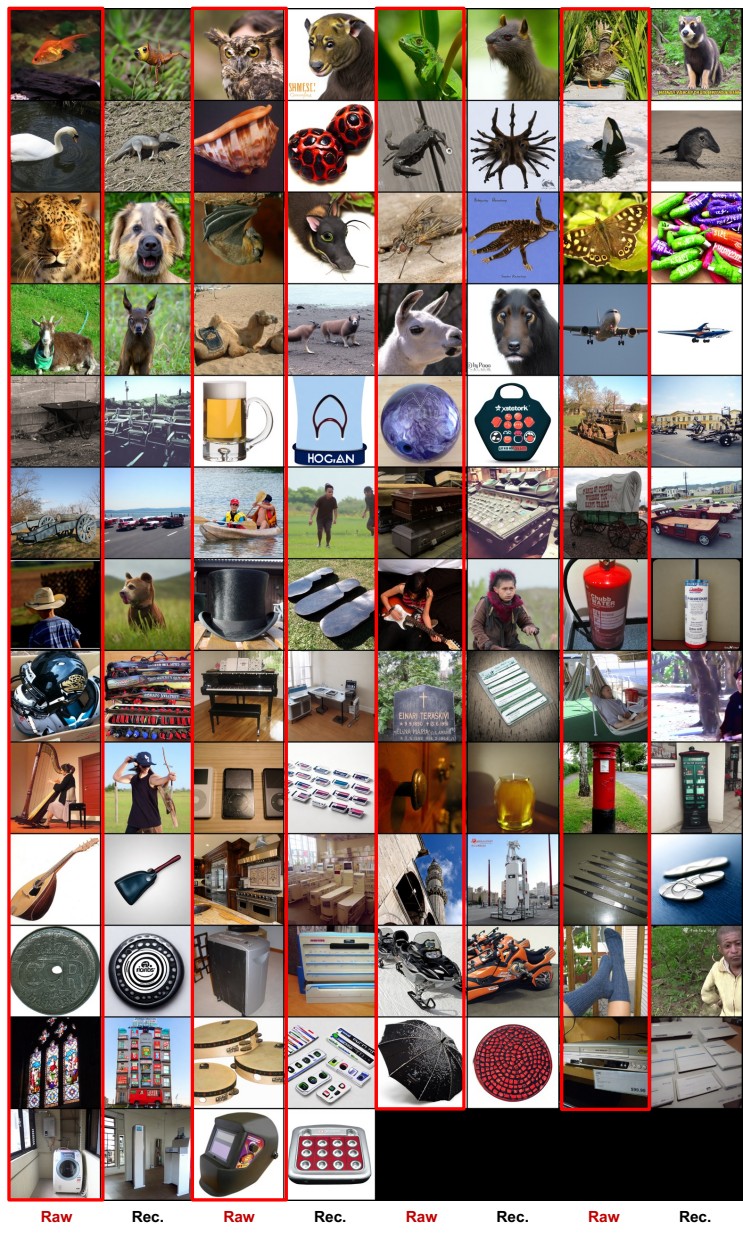

**Raw** Rec. **Raw** Rec. **Raw** Rec. **Raw** Rec.

Figure 7: Fully sampled and reconstructed images of subject 2.

Table 5: Effectiveness of different modules by perceptual similarity (CLIP).

| Subject | Sub1 (%) | Sub2 (%) | Sub3 (%) | AVG (%) |
|---|---|---|---|---|
| FULL method | 78.0±0.7 | 84.8±0.8 | 80.4±0.6 | 81.1±0.7 |
| w.o.MOE | 69.2±1.0 | 75.2±3.0 | 77.2±1.0 | 73.9±1.7 |
| w.o. momentum alignment | 63.6±0.1 | 60.2±2.7 | 68.2±0.4 | 64.0±1.1 |
| w.o. data augmentation | 72.4±3.0 | 76.4±0.6 | 78.2±2.4 | 75.7±2.0 |
| w.o. CycleGAN feat. | 75.6±1.8 | 78.0±0.8 | 78.0±0.3 | 77.2±1.0 |
| w.o. linear reprojection | 68.4±1.4 | 70.8±2.7 | 76.2±1.0 | 71.8±1.7 |

[2] Roman Beliy, Guy Gaziv, Assaf Hoogi, Francesca Strappini, Tal Golan, and Michal Irani. From voxels to pixels and back: Self-supervision in natural-image reconstruction from fmri. In *Advances in Neural Information Processing Systems*, pages 6514–6524, 2019.

[3] Christopher M Bishop and Nasser M Nasrabadi. *Pattern recognition and machine learning*, volume 4. Springer, 2006.

[4] Shu-Yu Chen, Wanchao Su, Lin Gao, Shihong Xia, and Hongbo Fu. Deepfacedrawing: Deep generation of face images from sketches. *ACM Transactions on Graphics (TOG)*, 39(4):72–1, 2020.

[5] Zijiao Chen, Jiaxin Qing, Tiange Xiang, Wan Lin Yue, and Juan Helen Zhou. Seeing beyond the brain: Conditional diffusion model with sparse masked modeling for vision decoding. *arXiv preprint arXiv:2211.06956*, 1(2):4, 2022.

[6] Anja Feldmann and Ward Whitt. Fitting mixtures of exponentials to long-tail distributions to analyze network performance models. *Performance evaluation*, 31(3-4):245–279, 1998.

[7] Matteo Ferrante, Tommaso Boccato, and Nicola Toschi. Semantic brain decoding: from fmri to conceptually similar image reconstruction of visual stimuli. *arXiv preprint arXiv:2212.06726*, 2022.

[8] Tomoyasu Horikawa and Yukiyasu Kamitani. Generic decoding of seen and imagined objects using hierarchical visual features. *Nature communications*, 8(1):15037, 2017.

[9] Furkan Ozcelik, Bhavin Choksi, Milad Mozafari, Leila Reddy, and Rufin VanRullen. Reconstruction of perceived images from fmri patterns and semantic brain exploration using instance-conditioned gans. In *2022 International Joint Conference on Neural Networks (IJCNN)*, pages 1–8. IEEE, 2022.

[10] Yu Takagi and Shinji Nishimoto. High-resolution image reconstruction with latent diffusion models from human brain activity. *bioRxiv*, pages 2022–11, 2022.

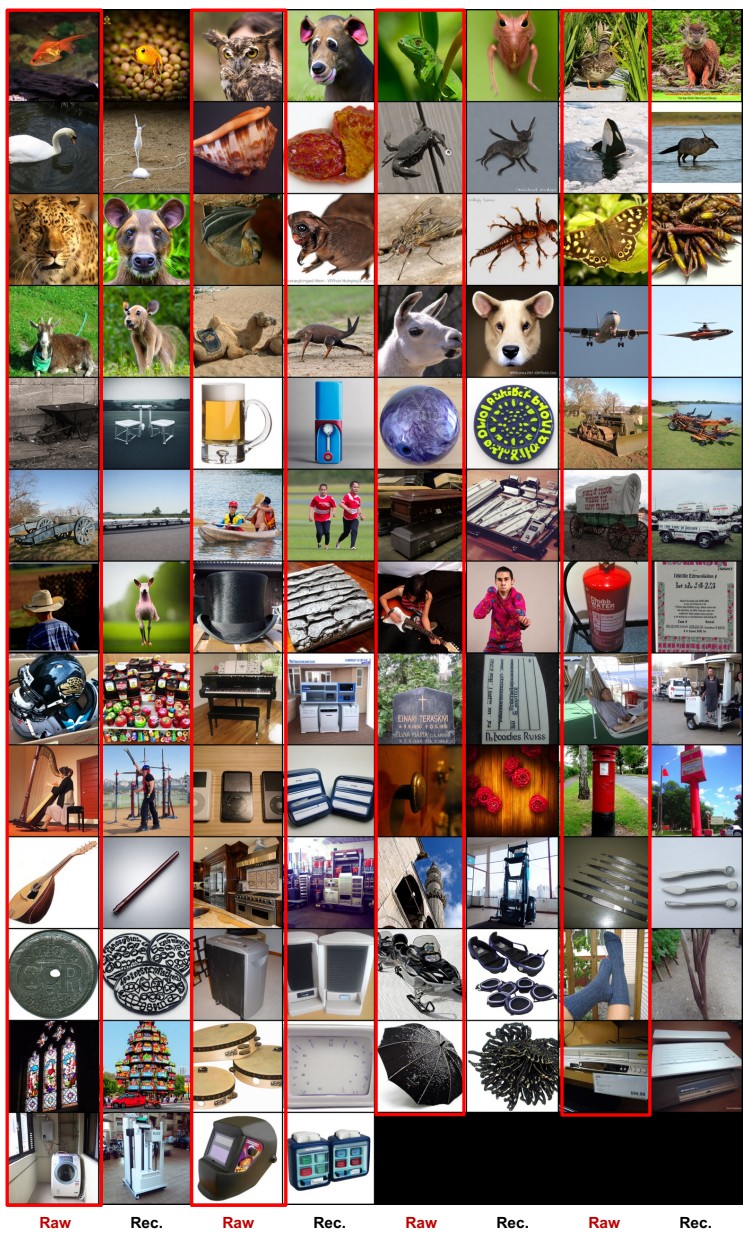

**Raw** Rec. **Raw** Rec. **Raw** Rec. **Raw** Rec.

Figure 8: Fully sampled and reconstructed images of subject 3.