# OpenReview forum: "Alleviating the Semantic Gap for Generalized fMRI-to-Image Reconstruction"
_NeurIPS.cc/2023/Conference — NeurIPS 2023 spotlight_

### Official Review · Reviewer_AK4m · 2023-06-30

**Soundness:** 3 good
**Presentation:** 4 excellent
**Contribution:** 3 good
**Rating:** 6
**Confidence:** 3

**Summary:**

This paper presents a new approach to generalized fMRI-to-image reconstruction with a focus in incorporating mage semantics and addressing semantic gaps during the reconstruction. To address inside-space semantic gap, a CLIP based feature space is utilized. To address outside-space semantic gap, a structural information guided diffusion model to transfer semantics. An adaptive strategy to integrate the semantic and structural information is also used. Experiments are conducted on the GOD and NSD dataset with several existing baselines.


**Strengths:**

The paper is well motivated and the methodology components clearly described. The focus on incorporating semantics and addressing semantic gaps is interesting, and the adaptive integration with LDM is novel.

The experiments are relatively comprehensive, and the margins of improvements especially on GOD appears to be significant.

The ablation study provides clear evidence on each module of the methodology, especially their adaptive integration.


**Weaknesses:**

It is not clear why did the authors consider only one baseline for comparison on NSD? The performance metrics only lack sufficient statistics on this dataset.

It is not clear if the training and testing is done separately per subject, or for all three subjects simultaneously but without held-out subjects?

**Questions:**

Please respond to the questions raised above.

**Limitations:**

The authors discussed briefly some limitations associated with the current methodology, without discussing potential future directions to address such limitations or their impact.

---

> ### Author Rebuttal · Authors · 2023-08-08
>
> ## To Reviewer AK4m
> ### W1. About only considering one baseline model for comparison on NSD.
> Given that in Section 4.5, we want to illustrate that following the dataset split method in [a], even a very simple baseline model (k-nearest-neighbor) can achieve a comparatively good decoding result. It could indicate that such a random split cannot ensure that the model has learned the visual representation mechanism and maintain generality. Therefore, we propose a more realistic zero-shot learning (ZSL) division method. The aforementioned baseline methods fail to generalize (Figure 4, 52% in ZSL split), but GESS can generalize well in this complex scenario.
>
> For sufficient statistics, we provide variance across repeated experiments (10 trials with different seeds) in Table R2 and will provide more details of the experiments in the supplemental materials.
>
> ### W2. Details of the training and testing split.
>
> Following [b], we train a model on one subject's training trials (1200 samples from 150 categories) and test it on the test trials (50 samples from 50 categories) of the same subject. The training and testing images come from different categories to construct a semantic gap (or ZSL scenario). We repeat our experiments on three subjects to demonstrate the generality and average the performance for the final results.
>
> Cross-subject generalization remains challenging. Because brain signals are highly personalized with different signal dimensions and visual area locations across subjects [c], effectively aligning data from multiple subjects into a shared space is difficult. Addressing cross-subject decoding is thus important but complex, beyond the scope of the current work. We leave exploring methods to achieve cross-subject prediction for future work.
>
>
> ### Limitations. Discussions of future directions to address the limitations.
>
> (1) About low SNR in fMRI, combining EEG [d] and fMRI [b] may capture better brain signals and provide complementary information, yielding more robust and accurate models for decoding semantic representations. Integrating multi-modal neural data has the potential to significantly advance performance.
>
> (2) Cross-subject generalization is an important next step. Collecting experimental recordings is expensive, and the current models [a] require larger datasets, so that training a model on several subjects' neural signals could provide further gains in accuracy.
>
> (3) More efficient inference strategy. The current model samples one image by multiple time steps which has greater computational cost compared to GANs. With the development of LDM, a more efficient solver is expected.
>
> We will add more future direction descriptions in the paper. All of the details mentioned above will be added to the paper or supplementary material.
>
>
> ### Table R2: Effectiveness of different modules by perceptual similarity (CLIP).
> | Subject | Sub1 (%) | Sub2 (%) | Sub3 (%) | AVG (%) |
> |---------|---------:|---------:|---------:|--------:|
> | Full Method | 78.0±0.7 | 84.8±0.8 | 80.4±0.6 | 81.1±0.7 |
> | w.o.MOE | 69.2±1.0 | 75.2±3.0 | 77.2±1.0 | 73.9±1.7 |
> | w.o. momentum alignment | 63.6±0.1 | 60.2±2.7 | 68.2±0.4 | 64.0±1.1 |
> | w.o. data augmentation | 72.4±3.0 | 76.4±0.6 | 78.2±2.4 | 75.7±2.0 |
> | w.o. CycleGAN feat. | 75.6±1.8 | 78.0±0.8 | 78.0±0.3 | 77.2±1.0 |
> | w.o. linear reprojection | 68.4±1.4 | 70.8±2.7 | 76.2±1.0 | 71.8±1.7 |
>
> ## References
> [a] Takagi, Yu, and Shinji Nishimoto. "High-resolution image reconstruction with latent diffusion models from human brain activity." Proceedings of the IEEE/CVF Conference on Computer Vision and Pattern Recognition. 2023.
>
> [b] Tomoyasu Horikawa and Yukiyasu Kamitani. Generic decoding of seen and imagined objects using hierarchical visual features. Nature communications, 8(1):15037, 2017.
>
> [c] Rieck, Bastian, et al. "Uncovering the topology of time-varying fMRI data using cubical persistence." Advances in neural information processing systems 33 (2020): 6900-6912.
>
> [d] Bai, Yunpeng, et al. "DreamDiffusion: Generating High-Quality Images from Brain EEG Signals." arXiv preprint arXiv:2306.16934 (2023).

---

### Official Review · Reviewer_qNjJ · 2023-07-04

**Soundness:** 3 good
**Presentation:** 3 good
**Contribution:** 3 good
**Rating:** 6
**Confidence:** 4

**Summary:**

This paper addresses the problems of semantic gap between training and testing fMRI neural responses and generalization of fMRI-to-image reconstruction models. A pre-trained CLIP model is leveraged to map the training data to a latent feature space in which sparse semantics are extended into dense semantics, thereby alleviating the semantic gap within known semantic spaces.  Overall, it is an interesting paper, and the empirical studies show some improvement.

**Strengths:**

Please refer to the question section

**Weaknesses:**

Please refer to the question section

**Questions:**

The followings are the major concerns and minor comments:


1) In this paper, the notations are confusing. In regular papers, scalers are denoted by small letters, vectors are defined with small letters (highlighted in bold), and matrices are denoted by capital letters using bold. In this paper, there are a lot of conflicts. It is so hard to trace what is a set, a matrix, or even a distribution.


2) The proposed method can be summarized in the form of an algorithm or pseudocode.


3) This paper is hard to follow. To explain the technical details of the proposed method clearly, some sections should be revised and reorganized.


4) Some abbreviations are presented before their definitions – e.g., CLIP in the abstract.


5) There are some minor linguistic and typo problems in this paper. E.g. “Alleviating” not, “Allievating” in the title of the paper.

**Limitations:**

Please refer to the question section

---

> ### Author Rebuttal · Authors · 2023-08-08
>
> ## To Reviewer qNjJ
> ### Q1. About the confusing Notations.
> We acknowledge the potential confusion caused by the notations employed in the paper and recognize the need for adopting clearer conventions. Accordingly, we will make revisions to define vectors with bold lowercase letters and represent matrices with uppercase letters using bold font, to achieve overall harmony in the final version. Additionally, for the sake of clarity (Algorithm R1), we have modified the subscript notation in the original text to superscript. Furthermore, we will ensure that the paper adheres to standard conventions.
>
> Concretely, (1) we will add subscripts to denote the conditional variable $h$ (in line 207) for clarity and provide more descriptions for each variable to make their meanings intuitive. (2) Some variables are noted with the subscript 'te' while others are not (like in line 235), depending on the context, which can be confusing. To standardize the notation, we will use superscripts uniformly across all variables. (3) We notice that in our paper, functions, modules, and matrices are all denoted using uppercase letters (e.g. lines 139, 171, 236, etc.), which is confusing. To follow standard conventions, we will revise the notations.
>
> In summary, we will make revisions to make the final version easier to follow for readers.
>
> ### Q2. To summarize as a Pseudo code description.
> We realize that presenting the proposed method in the form of pseudo code could aid clarity and we provide it in Algorithm R1. Kindly note that we have modified the subscript notation in the original text to superscript for clarity. We will include this description in the revised paper.
>
> ### Q3 About technical details for reproducibility.
>
> Due to page limits, some details were omitted or put in the supplementary materials like the detailed parametric settings (Section 3.4). To improve reproducibility and readability, we will add more details:
>
> (1) In Section 3.2.1, we will provide more context for the momentum alignment and linear reprojection methods. This includes the motivations, assumptions and parameter values.
>
> (2) The main text lacked details of the structural information extraction module, such as the transformer size [d] and codebook size. Section 3.2.2 will include more details on the feature extraction using CycleGAN [a].
>
> (3) Section 3.3 omitted some details on the normalization, kernel density estimation and mixture of experts methods, as well as parameters of our proposed conditioning strategy. We will expand this discussion in the main text and supplementary materials.
>
> (4) Section 4.1 lacked details on the dataset and preprocessing steps, which can be found in references [b][c]. We will add the necessary information in the supplementary materials.
>
> Later, we will provide intermediate features and additional results to enable reproducibility. We will also **release the code** to facilitate understanding of the work and enhance reproducibility.
> ### Q4. Q5. Some correctness.
> We acknowledge that some sections could be improved by correcting all the spelling errors in the revised version. We will also revise the case where abbreviations are presented before their definitions (line 45 for CLIP, line 117 for VQGAN, and some others in line 71, etc.). Additionally, we will correct spelling errors such as "alleviating", and address inaccuracies in word choices, such as revising "sub-class." These revisions will be made in the final version.
>
>
> Thank you for your advice and reminders. We will add more details and revise the paper accordingly to improve the reading experience.
>
> ## Algorithm R1
> Image reconstruction from fMRI using GESS (component constitution strategy).
>
> **Input**:
> Paired fMRI $X$ and Image $Y$ Dataset: $D^{tr} = {(x_i^{tr}, y_i^{tr})}^N_{i=1}$ and $D^{te} = {(x_i^{te})}_{i=1}^N$, N is the number of samples.
>
> **Output**: Reconstructed images $\hat{y}^{te}$ from fMRI $x^{te}$.
>
>
> **Training**:
>
> Initialization: Constructing CLIP [a], VQGAN [b] and LDM [c] by pretrained parameters $\phi$, $\gamma$ and $\theta$.
>
> Training $\beta_c$ of **semantic module $M_c$**:
> 1. Extracting semantic features $c_i^{tr}$ from $y_i^{tr}$ by CLIP: $c_i^{tr}=f_
> {\phi}(y_i^{tr})  $.
> 2. Training ridge regression parameters $\beta_c$ by {$c_i^{tr}, x_i^{tr}$.}
>
> Training $\beta_s$ of **structural model $M_s$**:
> 1. Extracting structural features $s_i^{tr}$ from $x_i^{tr}$ by VQGAN: $s_i^{tr}=f_
> {\gamma}(y_i^{tr})  $.
> 2. Flatting $s^{tr}_i$, and training ridge regression parameters $\beta_s$ by {$s_i^{tr}, x_i^{tr}$}.
>
>
> **Inference**:
> 1. **Semantic module.** Predicting semantics from fMRI: $\hat{c_i}^{te} = \beta_cx_i^{te}$. Using momentumn alignment and linear reprojection to get $\hat{c}^{te}_{i,r}$ (Section 3.2.1).
> 2. **Structural module.** Predicting structure from fMRI: $\hat{s}^{te}_i = \beta_sx_i^{te}$.
> 3. **MOE.** Estimating weighting parameters $\pi_c$ by KDE and $\pi_s = 1-\pi_c$ (Section 3.3.1).
> 4. **LDM.** Reconstructing by CS strategy (Section 3.3.2):
>     1. Initializing $z_0$ by Gaussian.
>     2. For $t = 1, 2, ..., T$:
>         1. Conditioned by cross attention: $z_{t-1}' \sim p_{\theta}(z_{t-1}'|z_t,\hat{c}^{te}_{i,r})$.
>         2. Conditioned by CS strategy: $z_{t-1}=z_{t-1}'-\pi_sF_s(z_{t-1}')+\pi_s F_s(\hat{s}^{te}_{i})$
>
>     3. $\hat{y_i}^{te} = f_{\gamma}(z_0)$.
>
> [a] Roman Beliy, Guy Gaziv, Assaf Hoogi, Francesca Strappini, Tal Golan, and Michal Irani. From voxels to pixels and back: Self-supervision in natural-image reconstruction from fmri.
>
> [b] Tomoyasu Horikawa and Yukiyasu Kamitani. Generic decoding of seen and imagined objects using hierarchical visual features.
>
> [c] Takagi, Yu, and Shinji Nishimoto. "High-resolution image reconstruction with latent diffusion models from human brain activity."
>
> [d] Patrick Esser, Robin Rombach, and Bjorn Ommer. Taming transformers for high-resolution image synthesis.

---

> ### Comment · Reviewer_qNjJ · 2023-08-20
>
> I have read all the reviews and rebuttals. I am satisfied with the author's responses to my concerns and still find the manuscript above the acceptance threshold. I raise my score to 6.

---

### Official Review · Reviewer_NuyD · 2023-07-05

**Soundness:** 3 good
**Presentation:** 2 fair
**Contribution:** 3 good
**Rating:** 6
**Confidence:** 4

**Summary:**

This paper proposes a GESS model to solve the semantic gap between the training and the testing data in the generalized fMRI-to-image reconstruction task. A CLIP based method is used to alleviate semantic gap for instances with known semantic space, and a structural information guided diffusion model is used to alleviate semantic gap for instances with unknown semantic space. In addition, this paper quantifies the semantic similarity between a given instance and training data.

**Strengths:**

Originality: The design of Generalized fMRI-to-image reconstruction task is interesting. It considers both the known and the unknown subspace, and proposes corresponding solutions.
Clarity：The motivation of the method is clearly addressed.
Significance: The proposed method not only explicitly extracts semantic and structural information, but also adaptively integrate the features based on the semantic uncertainty, alleviating the semantic gap and achieving general and vivid reconstructions.

**Weaknesses:**

1. The comparison is insufficient which maybe partially due to the specificity of the task.
2. The quantitative results are insufficient, and the ablation experiment is incomplete, which cannot well explain the quality of the proposed method. For example, "Momentum alignment" and "Linear reprojection" described in 3.2.1, and "VQ-GAN" and "CycleGAN" in 3.2.2, etc.
3. Quantified Semantic Confidence is not given in the experiment.

**Questions:**

It is confusing between "VQGAN to extract the latent representation" on line 190 and the decoding part of VQ-GANd in Figure 2. As far as I understand, this part of the figure uses VQ-GAN to encode features, and the encoding part should be used.
Is it possible to provide a computational efficiency comparison of the models?

**Limitations:**

It's better to include the discussions about limitations on experiment validation.

---

> ### Author Rebuttal · Authors · 2023-08-08
>
> ## To Reviewer NuyD
> ### W1. About the limited comparison experiments.
> When we submitted the paper, we strived to find limited open-source methods ([a], [b], [c], etc.), among which [c] is the state-of-the-art of CVPR 2023. We will continue searching and include more up-to-date methods for comparison.
>
> ### W2. More quantitative experiments of ablation studies.
> Due to space constraints, we did not include many ablation studies. To demonstrate the effectiveness of the proposed modules in Sections 3.2 and 3.3, we add more quantitative results (including the mean and standard deviation of the comparison metrics) and ablation studies (the effectiveness of momentum alignment, linear reprojection, CycleGAN features, data augmentation, MOE strategy, etc.) in Figure R1, R2 and Table R2.
>
> ### W3. About the quantified semantic confidence.
>
> In our approach, the semantic confidence is implicitly measured by kernel density estimation (KDE) in the MOE component. To quantify the semantic confidence, we perform a 100-class classification task and use the maximum of the estimated posterior probabilities as the instance confidence. As shown in Table R3, the semantics estimated by our model is comparatively confident across different subjects (93.6% on average).
>
> To further demonstrate the effectiveness of our estimated semantic confidence, we compared the performance of our model reconstructed with MOE-allocated weights (confidence) to those with a constant weight in Table R2. Results (73.9% vs 81.1% with MOE) show the benefit of our estimated semantic confidence.
>
> ### Q1. Details of VQGAN module.
> In our approach, we treat the VQGAN model as a perceptual compression method and use its encoder to extract compressed image features similar to the CNNs. The subsequent diffusion process and other calculations are all performed in the compressed image feature space. The VQGAN decoder is responsible for ultimately decoding the compressed features into the image space for reconstruction. We will add the above details to the paper to clarify our usage of VQGAN's encoder and decoder.
>
> ### Q2. About the computational efficiency comparison.
> We provide a list of the computational costs of the individual modules in our method and provide a computational cost comparison between our proposed GG and CS strategies in Table R1. The results demonstrate that the CS achieves approximately 3 times faster performance compared to GG. Further analysis indicates that the most computationally demanding components of the model are the inference and embedding stages. We will add the above results to the Supplementary Material.
> ### Limitations. More about the experiments.
> Regarding the experimental results, the current model limits its ability to generalize across subjects. As great difference exists across different subjects' signals, we cannot validate whether it can accurately decode semantic representations for new subjects.
>
> And following [b], we had to average the repeated fMRI recordings to improve their signal-to-noise ratio, which is an inefficient use of the available data. We will expand the discussion of these limitations and avenues for future improvement in the paper.
>
> ### Table R1: Time-consuming of each module.
> | Method | Time-consuming |
> |-|:-:|
> | Gradient Guided strategy | 679.87 s |
> | Component Substitution strategy | 215.45 s |
> | Fitting Semantic Module $M_c$| 0.158 s |
> | Fitting Structural Module $M_s$ | 2.459 s |
> | CLIP embedding (pre-processing) | 47.553 s |
> | VQVAE embedding (pre-processing) | 72.366 s |
>
>
> ### Table R2: Effectiveness of different modules by perceptual similarity (CLIP).
> | Subject | Sub1 (%) | Sub2 (%) | Sub3 (%) | AVG (%) |
> |---------|---------:|---------:|---------:|--------:|
> | Full Method | 78.0±0.7 | 84.8±0.8 | 80.4±0.6 | 81.1±0.7 |
> | w.o.MOE | 69.2±1.0 | 75.2±3.0 | 77.2±1.0 | 73.9±1.7 |
> | w.o. momentum alignment | 63.6±0.1 | 60.2±2.7 | 68.2±0.4 | 64.0±1.1 |
> | w.o. data augmentation | 72.4±3.0 | 76.4±0.6 | 78.2±2.4 | 75.7±2.0 |
> | w.o. CycleGAN feat. | 75.6±1.8 | 78.0±0.8 | 78.0±0.3 | 77.2±1.0 |
> | w.o. linear reprojection | 68.4±1.4 | 70.8±2.7 | 76.2±1.0 | 71.8±1.7 |
>
>
> ### Table R3: Averaged confidence across subjects on testset.
> | Subject | Confidence (%) |
> |-|:-:|
> | Subject1 | 93.2±14.2  |
> | Subject2 | 94.4±10.1 |
> | Subject3 | 93.3±12.3 |
> | Average | 93.6±12.2 |
>
> ## References
> [a] Roman Beliy, Guy Gaziv, Assaf Hoogi, Francesca Strappini, Tal Golan, and Michal Irani. From voxels to pixels and back: Self-supervision in natural-image reconstruction from fmri. In Advances in Neural Information Processing Systems, pages 6514–6524, 2019.
>
> [b] Tomoyasu Horikawa and Yukiyasu Kamitani. Generic decoding of seen and imagined objects using hierarchical visual features. Nature communications, 8(1):15037, 2017.
>
> [c] Zijiao Chen, Jiaxin Qing, Tiange Xiang, Wan Lin Yue, and Juan Helen Zhou. Seeing beyond the brain: Conditional diffusion model with sparse masked modeling for vision decoding. arXiv preprint arXiv:2211.06956, 1(2):4, 2022.

---

> > ### Comment · Reviewer_NuyD · 2023-08-21
> >
> > Thanks for providing the feedback. The responses have clarified some of my concerns and I would like to raise my score to weak accept.

---

### Official Review · Reviewer_ciXJ · 2023-07-07

**Soundness:** 3 good
**Presentation:** 2 fair
**Contribution:** 3 good
**Rating:** 7
**Confidence:** 3

**Summary:**

This paper's objective is to enhance the generalization performance of the fMRI-to-image reconstruction task through dense representation learning. To achieve this, a pre-trained CLIP is utilized to establish a semantic space, thereby bridging the gap between the training and test sets. Specifically, the paper presents an adaptive method for integrating semantic and structural information. A latent diffusion model is developed to align the semantic and structural data using the proposed gradient-guided method. Finally, the method's effectiveness is evaluated using both GOD and NSD datasets.

**Strengths:**

The idea of the paper looks novel and easy to read and the proposed method looks novel.


**Weaknesses:**

While I recognize that the paper presents a straightforward and novel extension, I am not completely persuaded by its benefits.

In its current state, the manuscript lacks clarity regarding the distinct advantages of each section.

Furthermore, I believe additional comparisons with other image reconstruction methods would be beneficial to fully substantiate the paper's contributions.

**Questions:**

While I appreciate the use of data augmentation for regularization in Momentum alignment, could you clarify how this might be viewed as a novelty of the paper in addressing the semantic domain shift?

Could you provide more insight into the assumption of a linearly weighted sum of neighboring elements mentioned in line 169? Why is this assumption necessary?

Regarding line 240, the authors examine the computational complexity of GG. It seems that ablation studies both in terms of computational and generalization performance would be beneficial.

The discussion on the outside-space gap is interesting, yet I feel some experimental evaluations demonstrating this problem in real-world cases would strengthen the argument.

In line 282, the paper considers the performance of other methods using the paper's new split strategy. Why should a new training and test split cause a decrease in performance? What would the method's accuracy be under the previous split? I believe a more comprehensive comparison is needed for a fair assessment.

And finally, what about between-subject accuracy? Do you think that the current method can manage the domain shift and perform well in between-subject contexts?


**Limitations:**

Indeed, the paper discusses the issue of significant variance in generation quality when dealing with noisy data. Could you spell please check “Allievating” in the title.

---

> ### Author Rebuttal · Authors · 2023-08-08
>
> ## To Reviewer ciXJ
> ### Q1. About the novelty of data augmentation in momentum alignment
> Different from previous works [b][d], we explicitly define the semantic gap and reduce it through momentum alignment. The alignment requires accurate descriptions of the data distribution while estimating statistics accurately from the limited fMRI data that deviates from real-world distributions remains a challenge. Therefore, the data augmentation used in momentum alignment is important for our novel contribution 1 (Section 1), as it helps learn well-aligned CLIP features. Ablation studies in Table R2 demonstrate its effectiveness (75.7% v.s. 81.1% of full method). Kindly note that our contribution 1 involves several components (momentum alignment, linear reprojection, etc.).
>
>
> ### Q2. About the importance of linear weighting assumption.
>
> $c_{te}$ is reprojected from $c_{te}^*$ as a linear combination of its nearest neighbors. Practically, we found that optimizing based on the MSE loss in Section 3.2.1 does not ensure that $c_{te}^*$ lies on the manifold, and this deviation leads to performance degradation (Table R2, 71.8% v.s. 81.1% of full method). To project the vector onto this unknown manifold, referring to [a], we make the assumption that after the aforementioned augmentation, the feature space $C_a$ is locally continuous such that a linear combination of its vectors approximately lies on the manifold. With this non-parametric assumption, we do not need to explicitly define the manifold or fit an additional model to approximate it, making our approach more efficient and effective.
>
> ### Q3. The computational complexity of GG and CS methods.
> To compare the computational costs of the two conditioning strategies, we measured the inference time costs of the GG and CS methods when generating 50 test images of size 768 x 768 pixels using 50 DDIM steps on an NVIDIA RTX 4090 GPU, AMD Ryzen 5950x CPU, and 64GB RAM. As shown in Table R1, the CS method requires approximately 3 times less time than the GG method, demonstrating significantly better computational efficiency.
>
> ### Q4. More experiments and examples of outside-space cases.
> In our model, the outside-space problem is addressed by the MOE strategy (Section 3.3.1). As shown in Table R2, when outside-space examples are not explicitly handled (only constant weighting assumption), performance degrades significantly (73.9% vs 81.1% of the full method). Concretely, in the third row of plots of Figure 4, when the building photos' concept is absent from the training set, the baseline model predicts according to prior experience and generates an irrelevant reconstruction (fruit), while GESS, which considers the zero-shot learning (ZSL) scenario, works comparatively well.
>
> ### Q5. More discussions and comprehensive comparison of split strategy.
> The generalized split in our paper differs from the random split in that our split strategy considers the zero-shot learning (ZSL) scenario, where the training and test sets come from different categories. Such split follows [b] and aligns with reality: in experiments, collecting brain signals is expensive and time-consuming, which leads to a limited sampling scope.
>
> When a random split is considered (the training and test sets have high semantic overlap), both our model and the baseline model perform well (Figure 4, 81% vs 88.8% accuracy by perceptual similarity).
>
> Regarding the performance degradation (81% v.s. 52% of baseline model, in Figure 4), some methods that are trained on a limited number of samples tend to be overfitted without understanding the underlying visual mechanisms. As a result, they fail to generalize to real-world images under the more challenging ZSL split.
>
>
> ### Q6. About applying GESS in the cross-subject case.
> Generalizing GESS to the cross-subject scenario remains challenging. The brain signals are highly individualised, with different signal dimension and visual area locations across subjects [b][c]. So that it is difficult for a shared network to process them directly. Addressing the lack of cross-subject generalization is complex and beyond the scope of the current work. We leave exploring methods to achieve cross-subject decoding for future work.
>
> ### Table R1: Time-consuming of each module.
> | Method | Time-consuming |
> |-|:-:|
> | Gradient Guided strategy | 679.87 s |
> | Component Substitution strategy | 215.45 s |
> | Fitting Semantic Module $M_c$| 0.158 s |
> | Fitting Structural Module $M_s$ | 2.459 s |
> | CLIP embedding (pre-processing) | 47.553 s |
> | VQVAE embedding (pre-processing) | 72.366 s |
>
> ### Table R2: Effectiveness of different modules by perceptual similarity (CLIP).
>
> | Subject | Sub1 (%) | Sub2 (%) | Sub3 (%) | AVG (%) |
> |---------|---------:|---------:|---------:|--------:|
> | Full Method | 78.0±0.7 | 84.8±0.8 | 80.4±0.6 | 81.1±0.7 |
> | w.o.MOE | 69.2±1.0 | 75.2±3.0 | 77.2±1.0 | 73.9±1.7 |
> | w.o. momentum alignment | 63.6±0.1 | 60.2±2.7 | 68.2±0.4 | 64.0±1.1 |
> | w.o. data augmentation | 72.4±3.0 | 76.4±0.6 | 78.2±2.4 | 75.7±2.0 |
> | w.o. CycleGAN feat. | 75.6±1.8 | 78.0±0.8 | 78.0±0.3 | 77.2±1.0 |
> | w.o. linear reprojection | 68.4±1.4 | 70.8±2.7 | 76.2±1.0 | 71.8±1.7 |
>
>
> ## References
> [a] Shu-Yu Chen, Wanchao Su, Lin Gao, Shihong Xia, and Hongbo Fu. Deepfacedrawing: Deep generation of face images from sketches. ACM Transactions on Graphics (TOG), 39(4):72–1, 2020
>
> [b] Tomoyasu Horikawa and Yukiyasu Kamitani. Generic decoding of seen and imagined objects using hierarchical visual features. Nature communications, 8(1):15037, 2017.
>
> [c] Rieck, Bastian, et al. "Uncovering the topology of time-varying fMRI data using cubical persistence." Advances in neural information processing systems 33 (2020): 6900-6912.
>
> [d] Takagi, Yu, and Shinji Nishimoto. "High-resolution image reconstruction with latent diffusion models from human brain activity." Proceedings of the IEEE/CVF Conference on Computer Vision and Pattern Recognition. 2023.

---

> > ### Comment · Reviewer_ciXJ · 2023-08-18
> >
> > I'm grateful for the author's thoughtful responses to my feedback. As a result, I've upgraded my rating from 'Weak Accept' to 'Accept'. I would be glad to see the revised version of the paper.

---

### Author Rebuttal · Authors · 2023-08-08

## To Reviewers
We sincerely appreciate all reviewers devoting time for our paper and provide valuable comments. We also feel encouraging that all reviewers agree with our contributions in addressing the semantic gaps, introducing the adaptive confidence-weighted approach, and presenting the specially designed condition strategy for the diffusion process.

We have taken meticulous care in addressing each of the concerns raised by the reviewers. Our responses have been crafted to effectively address the questions and provide comprehensive explanations.

Furthermore, we have included a PDF file that contains additional results (Figure R1 - R2) to substantiate our responses, particularly for aspects of additional ablation studies.


We will merge the details and figures from our responses into both the main text and the supplementary materials. Once again, we extend our gratitude for your valuable feedback, and we firmly believe that these refinements will significantly enhance the quality and impact of our work.

---

### Decision · Program_Chairs · 2023-09-21

**Decision:**

Accept (spotlight)

**Comment:**

All reviewers unanimously agree that there are significant merits to this paper. The paper presents an original and well-motivated design for a Generalized fMRI-to-image reconstruction task that takes into account both known and unknown subspaces, offering innovative solutions. The clarity of the paper is evident, as the motivation and methodology components are clearly described. Its significance lies in its unique approach to extracting semantic and structural information while adaptively integrating features to bridge the semantic gap, resulting in vivid reconstructions. This focus on semantics and adaptive integration is particularly novel. The experiments conducted are comprehensive, showing significant improvements.